# A Review of Additive Manufacturing Studies for Producing Customized Ankle-Foot Orthoses

**DOI:** 10.3390/bioengineering9060249

**Published:** 2022-06-09

**Authors:** Rui Silva, António Veloso, Nuno Alves, Cristiana Fernandes, Pedro Morouço

**Affiliations:** 1CIPER, Faculdade de Motricidade Humana, Universidade de Lisboa, 1495 Cruz Quebrada Dafundo, 1649-004 Lisbon, Portugal; rui.d.silva@ipleiria.pt (R.S.); apveloso@fmh.ulisboa.pt (A.V.); 2CDRSP, Polytechnic of Leiria, 2430-028 Marinha Grande, Portugal; nuno.alves@ipleiria.pt (N.A.); cristiana.fernandes@ipleiria.pt (C.F.); 3ESECS, Polytechnic of Leiria, 2411-901 Leiria, Portugal; 4ciTechCare, Center for Innovative Care and Health Technology, Polytechnic of Leiria, 2411-901 Leiria, Portugal

**Keywords:** lower extremity, rehabilitation, walking, customization, patient-specific

## Abstract

Ankle-foot orthoses (AFO) are prescribed to improve the patient’s quality of life. Supporting weak muscles or restraining spastic muscles leads to smoother and more stable locomotion. Commonly, AFO are made using thermoplastic vacuum forming, which requires a long time for production and has limited design options. Additive manufacturing (AM) can solve this problem, leading to a faster and cheaper solution. This review aimed to investigate what is the state-of-art using AM for AFO. Evaluating the used manufacturing processes, customization steps, mechanical properties, and biomechanical features in humans would provide significant insights for further research. The database searches combined AM and AFO with no year or publication type restrictions. Studies must have examined outcomes on human participants with the orthoses built by AM. Other types of orthotic devices or different manufacturing techniques were excluded. Nineteen studies met the inclusion criteria. As stated by having all studies conducted in the last nine years, this is a very recent domain. Different AM processes have been used, with the majority relying on Fused Deposition Modeling. Overall, the manuscripts’ quality is deficient, which is critical to promoting further studies with higher samples. Except for one paper, AM-printed AFO was comparable or superior to the thermoplastic vacuum forming AFO in mechanical tests, kinematics, kinetics, and participant feedback.

## 1. Introduction

Walking is one of the most critical events in daily living, and difficulty in walking is a substantial barrier for both adults and children [1]. Accordingly, ankle-foot orthoses (AFO) are prescribed to improve the patient’s quality of life for several walking difficulties. It is well documented that these devices may help in lower limb impairments such as stride length [2]; gait speed and walking confidence [3,4,5,6,7]; equinus ankle correction [2,8]; energy expenditure index [6]; hip extension, dorsiflexion in the swing phase and knee extension [2,3]; correction of knee hyperextension [9]; correction of foot drop [10]; correction of the crouch gait [8]; increased solear muscle activity [11]; and increased resistive moment in plantar flexion [9]. An AFO can support weak muscles or restrain spastic muscles, leading to smoother and more stable locomotion.

Today, patients can choose between standard off-the-shelf AFO and custom-made AFO. The former is cheaper but might offer less comfort to a patient than a custom-made AFO. On the other hand, a custom-made AFO may increase that comfort and be adequate, but the manufacturing process is far from optimal. The most common procedure to fabricate this type of AFO is mold. However, it takes a long time to make the mold and get the final product, which may take from two days to several weeks depending on the post-processing needed. Also, the technician needs to spend most of that time working on the orthosis, taking the time away from the work with patients and other aspects of their work [12]. Additionally, it is not adaptable to morphologic modifications (e.g., rapid body changes during children’s growth), requiring highly skilled personnel [13]. These inconvenient features illustrate how much research is necessary on this topic. For instance, if society can conceive a faster and cheaper method, it may be easier to change AFO along with the children’s growth.

Currently, there is no doubt that the massive customization of products and services is a regular trend over massive production, aiming for custom mass production [14]. With additive manufacturing (AM) being a little-explored domain, this technology allows for the customizing of a product since it is manufactured layer-by-layer, thus allowing complex architectures and formats [14]. These architectures are previously modeled in a virtual environment with computer-aided design (CAD) software, which differs from traditional production processes based on removing material or the deposition of materials in molds. Customization is essential for specific biomedical applications, such as orthopedics or orthotics, in which the efficiency of treatment is strongly connected with each patient’s anatomical geometry [15,16].

To the best of our knowledge, using AM for AFO production is a recent field of research. Thus, examining the available studies, their advantages, and drawbacks may provide significant further investigation insights. This review aimed to investigate the use of AM for AFO, exploring the manufacturing and customization processes and evaluating their mechanical and biomechanical properties.

## 2. Materials and Methods

Database searches were performed between October 2021 and January 2022 in Web of Science, SCOPUS, PubMed (including MEDLINE), and Scielo. Terms related to additive manufacturing (3D printing, additive manufacturing, selective laser sintering, fused deposition, rapid prototyping) combined with terms to AFO (ankle-foot, orthoses, orthosis) were used, without restrictions.

Original papers were written in English with ankle-foot orthoses developed by additive manufacturing, and human participants were included. Any sample size was eligible, and there were no restrictions on the type of participants (sex, age, culture, ethnicity, healthy, non-healthy). We have included additive manufacturing types (e.g., fused deposition modeling, selective laser sintering or melting, stereolithography, and digital light processing). The articles must have any outcomes by tests performed on human participants with the orthoses built by AM.

All narrative or systematic reviews were excluded, although the reference list was examined for additional references. Any full article not written in English or unpublished data were excluded. Any article with other types of orthotic devices (e.g., foot orthosis (F.O.), knee-ankle foot orthosis (KAFO, splint), or different manufacturing techniques (e.g., mold filling) were excluded.

Data extraction was standardized after removing the excluded articles and deleting duplicates. Titles and abstracts from the search results were screened using the eligibility criteria and reviewed by two authors (R.S. and P.M.) for inclusion. We have assessed the overall quality of evidence using the Grading of Recommendations Assessment, Development and Evaluation (GRADE) process (GRADEpro GDT) [17].

## 3. Results

Figure 1 illustrates the steps to identify relevant articles for the review based on PRISMA guidelines [18]. The initial database search identified 1466 articles, and after duplicate removal, 540 were considered potentially related and were screened for relevant content. No additional articles were identified following a hand search of reference lists. After reading the title and abstract of the 540 articles, 63 were selected for possible inclusion in this review, and full-text articles were retrieved. 19 of the 63 articles were included in this review in the last phase because they met the inclusion criteria. The 19 studies included outcomes such as mechanical tests [15,19,20,21,22], finite element method (FEM) simulations [19,21,23], participant feedback (healthy participants) [24,25,26] patient feedback (non-healthy participants) [19,22,27], QUEST [15,28], kinematics [15,22,23,26,28,29,30,31,32,33,34,35], kinetics [23,26,30,31,32,35], observation after trial [27,36], dimensional accuracy [25] and EMG [30,34,35].

A description of the AM AFO details can be found in Table 1. We have used the GRADE process to analyze the quality of the included studies (Table 2). The outcomes included were: (1) walking ability through biomechanical tests (kinematics, kinetics, EMG); (2) durability through mechanical test; (3) durability through observation after trial; (4) patient satisfaction assessed with the Quebec User Evaluation of Satisfaction with assistive technology (QUEST); (5) comfort through participant/patient feedback; (6) dimensional accuracy and material strength and AFO behavior simulation assessed by FEM analysis. All the outcomes obtained very-low quality evidence overall. 

We have compared the studies that used kinematics as an outcome with the data on the leg’s ankle and knee angles with the AM AFO in the stance phase (Table 3). The maximum angle for ankle dorsiflexion and knee flexion was 22° and 20°, respectively.

## 4. Discussion

Additive manufacturing methods to build ankle-foot orthoses are still in a very embryonic state, as shown by the papers’ publication date. All articles reported in this review have been carried out in the past nine years, and exponential growth is expected in the next decade with the evolution of additive manufacturing printers and the type of materials used. From the nineteen studies retrieved, just seven compared the customized AM AFO with the traditional thermoformed polypropylene AFO. Similar results in biomechanical tests and comfort were observed. Accordingly, the adoption of AM may lead to faster and cheaper processes having at least the same outcomes.

Researchers have been using different types of AM printing and materials. The majority of the papers used fused deposition modeling (FDM) [15,19,20,21,24,27,28,33,34] and selective laser sintering (SLS) [25,29,30,31,32,35,36]. Multi-jet fusion (MJF) [22] and stereolithography (SLA) [26] were also used, and one manuscript did not describe the printing method [23]. The AM printing method will bring pros and cons to the orthoses manufacturing and quality. The main advantage of the FDM process is that no chemical post-processing is required. No resins are necessary to cure; less expensive machines and materials lead to a more cost-effective process. Nevertheless, the resolution on the z-axis is lower than in other additive manufacturing processes [37], and inter-layer distortion was the leading cause of mechanical weakness [37]. Four of the eight studies that used FDM did some mechanical tests using acrylonitrile butadiene styrene (ABS), poly-lactic acid (PLA), and thermoplastic polyurethane (TPU) materials. Belokar et al. [20] showed that an ABS AFO could support a load of 10 tons, and the customized TPU AFO of Cha et al. [15] survived 300,000 repetitions in a durability test and two months of use by a foot drop in a 67-year-old patient. Although the customized PLA AFO of Maso and Cosmi [19] was considered excellent for manufacturing, it was not the most mechanically resistant.

Seven studies used the SLS printing process. Five studies used this process to build a complete AFO made of nylon 11 [25] and polyamide (Nylon) 12 (PA12) [29,31,32,36]. Two studies used SLS to manufacture a strut to change the stiffness of a pre-built carbon AFO made by the traditional method. SLS is a process in which a powder is sintered or fused by applying a carbon dioxide laser beam. The chamber is heated to almost the melting point of the material. The laser fuses the powder at a specific location for each layer specified by the design [38]. This technology’s main advantages are the wide range of materials used; however, in these studies, they just used Polyamide (Nylon) 12 and Nylon 11, which show almost the same mechanical properties as the injected parts [39]. The disadvantages are that the accuracy is limited by the size of particles of the material [38], the slow process, the high costs, and the high porosity when the powder is fused with a binder [40]. Although seven studies manufactured SLS AFO, no mechanical tests were made, and just one (Deckers et al. [36]) did an observation in children and adults with mixed results. Five did not survive the six-week trial of the seven built SLS AFO (calf and foot connected by two carbon fiber rods to change the stiffness). Three broke when doing sports (hiking, running, soccer), one broke while the patient walked upstairs, and one broke due to a manufacturing defect. Two survived the six weeks; nevertheless, one became dirty, and a cracking began at the metatarsal phalangeal joint. Telfer et al. [32] attached off-the-shelf gas springs with AM printed components (shank, strut, slider, and foot), allowing the user to change the stiffness of the AFO that could improve the ankle biomechanics helping day-to-day tasks reducing pain and fatigue. The results suggest that these devices may show equivalence in clinical performance compared with traditional AFOs, however, their mechanical performance is far from ideal. Yet, no comparison was made using unhealthy participants or traditional AFOs. 

Two studies used different printing techniques (SLA and MJF). SLA, which was developed in 1986, is one of the earliest additive manufacturing methods, and uses a liquid-based process that consists of the curing or solidification of a photosensitive polymer when an ultraviolet (UV) laser contacts the resin [38]. SLA prints high-quality parts at a fine resolution as low as 10 μm. However, it is relatively slow and expensive, the range of printing materials is minimal, it is sensitive to long exposure to UV light and the printed parts are affected by moisture, heat and chemicals. [38,40]. Mavroidis et al. used the SLA process with Acura 40 Resin and a DSM Somos 9120 Epoxy Photopolymer. No mechanical test was done. They achieved an optimal fit of the AM AFO geometry to the participant’s anatomy and achieved excellent comfort, and the AM AFO performed similarly to the traditional AFO. MJF combines SLS and binder jetting technologies. Compared to other AM methods, MJF has the lowest cost of 3D printed parts, quick printing, and no need for support; however, it is limited to just two types of material, and the machines are large and expensive [38,40]. For instance, a single unit of material for MJF may be up to four times less expensive than for FDM. Liu et al. [22] used the MJF process with Polyamide 12 material in stroke patients. The mechanical tests of the AFO showed toughness and high strength. They achieved a lightweight and comfortable AFO for the patient; however, further large-scale stroke samples and a long-term follow-up would be warranted to prove that MJF with PA12 could be a future solution to manufacturing custom AFO. Although different studies had utterly different methodologies and samples, the ABS and MJF AFOs obtained better durability results than the AFOs manufactured by SLS.

A GRADE evidence profile was created to assess the different outcomes in the included studies. The results analyzed had severe problems, mainly because most of them did not compare the created AM AFO with a traditional polypropylene AFO. Moreover, the number of participants/patients assessed was low. The outcomes from the included studies were very heterogeneous. Although some studies (*n* = 12) had kinematics in their results, they commonly used only the ankle (*n* = 10) and knee (*n* = 8) degrees. The lack of other critical kinematic variables in most of the studies (e.g., cadence, angular velocity, hip angle, gait speed, step length, stride length, duration of stance/swing), combined with the heterogeneity in the methodology, type of patients (the kind of disease, gender, and age) and different AM AFO makes it challenging to have a reliable quantitative comparison. In the future, it is believed that because it is an area with massive potential for expansion, studies will begin to have a greater homogeneity in their methodologies.

AFO users have different ages, anatomy, gender, and lifestyles, and can be found at various stages of the disease or disability. Stroke [41,42], multiple sclerosis [43], cerebral palsy [3,44,45], foot drop [2,8,46], Charcot-Marie tooth [47], neck or spinal cord injury [48], sciatica [44], muscular dystrophy [49], or peroneal nerve injury [46] are the most common diseases that need an AFO to improve the kinematics and kinetics of the patients. Among the AFO functionality, the patient’s comfort, pain, and disability reduction should be an essential factor to consider. In general, the reviewed papers present several flaws in their methodology. Of the studies, just six gave patient feedback for comfort and fit, and only two collected a QUEST. One study [20] presented interesting mechanical test results; however, no results were shown regarding the durability of the AFO after being applied to an end-user. Almost 50% of the studies presented in this review used healthy participants. While it is the easiest solution to test durability, comfort, uneven pressure distribution, redness, abrasions, or geometry to the participant anatomy, measuring its impact on groups with diseases is critical. Currently, the time from the prescription to the design of traditional polypropylene AFO can take several weeks, making them often unusable due to the constant changes in anatomy, particularly in children. Custom AM AFOs could have an essential role in solving the manufacturing time (less than one day), as shown by the two studies using children as participants [15,36]. Together with the manufacturing time, the capacity to create complex structures could be the solution to change the aesthetics of traditional AFOs, since some of the patients who need an AFO (mainly females and children) do not use them because of the appearance and finish of the orthosis [50].

Looking at all of the studies, further studies to build and test AM AFOs should include many more children and unhealthy participants. Furthermore, the studies should consist of all of these steps: (1) a 3D scan of the patient’s lower leg or plaster caster model; (2) CAD Modeling of the AFO for the patient condition; (3) FEM simulations to tune and predict the properties of the AFO; (4) AM printing of the AFO with the selected material; (5) Mechanical tests of the AFO; (6) biomechanical tests, durability, and satisfaction of the patient using the AFO.

The adoption of AM techniques for custom AFO may allow topological optimization, 4D manufacturing (manufacturing with smart materials), incorporation of multi-material leading to reduced weight and thickness, increased breathability, controlled flexibility, better fit, enhanced aesthetics, and the potential to eliminate several steps of production compared with traditional methods of AFO manufacture leading to a less cost and better AFO [13,51]. Furthermore, novel patient-specific AM AFO can substantially affect patient satisfaction, adherence to AFO usage, and overall health-related outcomes [50].

## 5. Conclusions

Nowadays, it is possible to manufacture a custom orthosis using AM. Nevertheless, it is far from becoming the ideal solution for clinical practice. The studies have shown that AM custom-made orthoses are comparable to the traditional AFO regarding kinematics, kinetics, and mechanics. In some cases, the AM custom-made orthoses performed better in comfort, performance, and optimal fit. However, the lack of more participants in studies with some diseases, the lack of more mechanical tests (e.g., durability and stiffness), no feedback from the participants, and the need for more tests of the pediatric population mean that additive manufactured orthoses have a way to go before they are used by the masses.

## Figures and Tables

**Figure 1 bioengineering-09-00249-f001:**
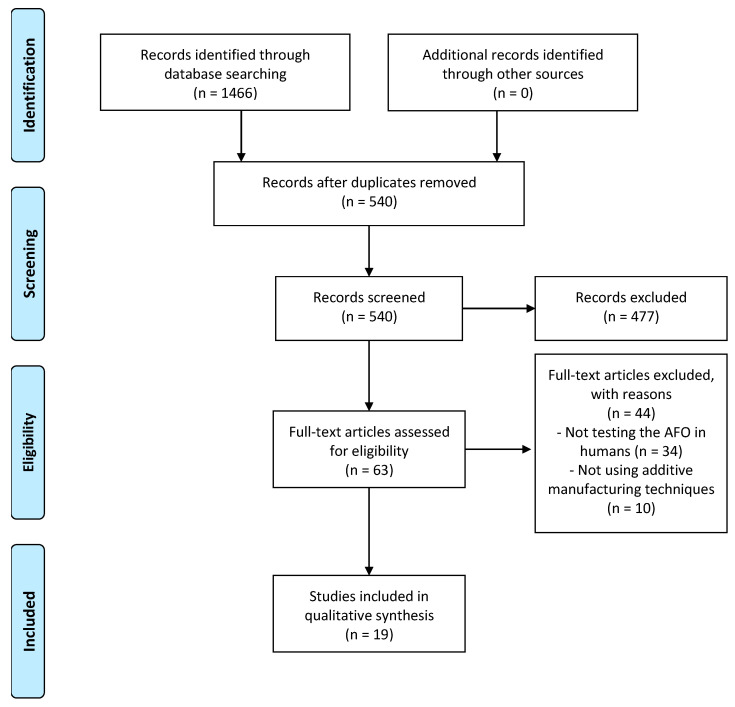
Flow diagram of the search history and selection process.

**Table 1 bioengineering-09-00249-t001:** Included studies with AFO details, participant/patient characteristics, intervention and control conditions, outcomes, and main results.

Reference	AFO Details		Participant/Patient Characteristics		Intervention vs. Control Condition	Outcomes	Main Results and Conclusions
	AM Printing Method	Material	N	Condition			
Belokar, Banga and Kumar, 2017 [20]	FDM	ABS	1 (M; 65 kg)	Healthy	Customized ABS AFO	Mechanical test	Maximum 6.8% strain with 38.4 MPa tensile strength exerted on the AFO. Rupture of the AFO at 14.96 kJ/m^2^ impact. No deformation in the inner surface with load up to 15 kN. No deformation of the AFO in hydraulic press test with 10 tons load.
Cha et al., 2017 [15]	FDM	TPU	1 (F; 68 years)	Foot drop on the right side after an embolectomy	Customized TPU AFO vs. TTPP AFO vs. Shod Only	Mechanical test; QUEST; kinematics	No structural change, crack or damage after 300k repetitions in the durability test. Both AFO increased gait speed and stride length. Step width decreased with the FDM AFO. Higher bilateral symmetry with FDM AFO induced more stability. Better satisfaction on the FDM AFO after using both AFO for 2 months.
Chae et al., 2020 [28]	FDM	TPU	1 (F; 72 years)	Foot drop on the right side after posterior lumbar interbody fusion and abscess	Customized TPU AFO vs. Without AFO	Kinematics; QUEST	Using the AFO, cardiorespiratory fitness and functionality increased. Stability score with eyes open and closed improved. In QUEST items, the device and service subscore had a perfect score (5 points) showing the patient’s satisfaction with the AFO.
Chen et al., 2014 [21]	FDM	ABS; ULTEM (Polyetherimide)	1 (M; 29 years; 68 kg)	Healthy	Customized ABS AFOs vs. TTPP AFO	Mechanical test; FEM simulations	The highest strains were found at about 50% of the gait cycle for PP (–15.3 × 10^−4^), ABS (–6.4 × 10^−4^), and ULTEM (–10.3 × 10^−4^). The FEM estimated rotational stiffness (N·m/deg) for PP (39.1), ABS (67.7) and ULTEM (89.0). Using calculated loading conditions and FEM can help design AFO to match the patient’s need and achieve desired biomechanical functions.
Choi et al., 2017 [34]	FDM	PLA	8 (4F; 4M; 25 ± 5 years; 1.7 ± 0.1 m; 67 ± 9 Kg)	Healthy	Customized PLA AFO with elastic polymer bands	Kinematics, ultrasound; EMG; musculoskeletal simulation	Use of elastic polymer bands to control the stiffness of the orthosis. More stiffness led to a decrease of peak in knee extension and ankle dorsiflexion angles and maximum length of the gastrocnemius and Achilles tendons. Due to medial gastrocnemius operating length and velocity changes, slower walking speeds may not receive the expected energy savings.
Creylman et al., 2013 [29]	SLS	Nylon 12 (PA2201)	8 (M; 47 ± 13 years; 1.97 ± 0.1m; 85.30 ± 14.20 Kg)	Unilateral Foot Drop due to dorsiflexor weakness	Customized Nylon 12 AFO vs. TTPP AFO vs. Bare Foot	Kinematics	Similar stride duration for all interventions. Significant differences in both AFO vs. barefoot for stride length of the affected (1377 vs. 1370 vs. 1213 mm) and unaffected (1373 vs. 1365 vs. 1223 mm) limb and stance phase duration of the affected limb (62.1 vs. 62.1 vs. 60.6%) for barefoot, AM AFO and TTPP. Range of Motion different between AFO due to Nylon 12 stiffer than PP.
Deckers et al., 2018 [36]	SLS	PA12	7 (4 Adults; 3 Children)	Trauma, Neuro-muscular disorder and cerebral palsy	Customized PA12 AFO with carbon fiber strut vs. TTPP AFO	Observation after trial	TTPP AFO (*n* = 7) survived the six weeks of clinical trial. For AM AFO (*n* = 7), three broke when doing sport, one broke while the patient walked upstairs, one broke due to a manufacturing defect, and one became dirty. A cracking began at the metatarsal phalangeal joint, and one survived with no problems.
Harper et al., 2014 [30]	SLS	Nylon 11 (PA D80—S.T.)	13 (M; 29 ± 6 years; 1.8 ± 0.1 m; 88 ± 11 Kg)	Unilateral lower extremity injuries	Customized Nylon 11 PD-AFO Strut (nominal vs. 20% stiffer vs. more compliant)	Kinematics; kinetics; EMG	Minimal effect in kinetics, kinematics and EMG gait cycle with different strut stiffness. Propulsive and medial GRF impulses were only influenced by AFO stiffness with the medial GRF impulse significantly increased in the stiff condition. Orthotists may not need to control the stiffness level precisely and may instead prescribe the AFO stiffness based on other factors.
Lin, Lin, and Chen, 2017 [33]	FDM	No Data	1	Healthy	Customized AFO vs. TTPP AFO	Kinematics	The walking speed (367 vs. 389 mm/s), stride length (583 vs. 598 mm), cadence (76 vs. 78 steps/min) and range of motion of knee joint in flexion were similar in both AFO. TTPP AFO obtained more extended range of motion due to different footplate.
Liu et al., 2019 [22]	MJF	PA12	12 (4F; 8M; 56 ± 9 years; 1.7 ± 0.1 m; 69 ± 10 Kg)	Stroke patients (6 Ischemic, 6 Hemorrhage).	Customized PA12 AFO vs. Without AFO	Mechanical test; kinematics; patient feedback	Using AM AFO increased velocity (0.17 ± 0.06 vs. 0.20 ± 0.07 m/s), stride length (0.43 ± 0.10 vs. 0.48 ± 0.11 m) and cadence (47.0 ± 14.4 vs. 53.8 ± 15.5 times/min). Double limb support phase (36.3 ± 5.6 vs. 33.6 ± 5.2 %) and the step length difference decreased (0.16 ± 0.12 vs. 0.10 ± 0.09 m). AM AFO obtained adequate dimensional accuracy, toughness, high strength, lightweight and comfort. No breakage occurred within three months.
Maso and Cosmi, 2019 [19]	FDM	PLA	1 (F; 21 years)	Post-traumatic rehabilitation	Customized PLA AFO	Mechanical Test; FEM simulations; patient feedback	Great geometrical correspondence and comfort between the foot and the AM AFO. Cheap production method compared with AFO produced with other technologies. PLA material was considered excellent for manufacturing the AFO but is not the most mechanically resistant.
Mavroidis et al., 2011 [26]	SLA	Accura 40 Resin; DSM Somos 9120 Epoxy Photopolymer	1	Healthy	Customized Accura 40 Resin AFO vs. Customized DSM Somos 9120 Epoxy Photopolymer vs. TTPP AFO vs. Shod only	Kinematics; kinetics; participant feedback	AM AFO obtained optimal fit and great comfort. Kinetics and Kinematics gait cycle revealed that the AM AFO performed similarly to the TTPP AFO.
Patar et al., 2012 [24]	FDM	ABS	1	Healthy	Customized ABS/PP DAFO (Dynamic Ankle-Foot Orthosis) vs. No control	Participant feedback	The price reduction in producing AM DAFO was reduced 100-fold compared to the products that existed in the market. The patient considered the performance was good.
Ranz et al., 2016 [35]	SLS	Nylon 11 (PA D80—S.T.)	13 (29.50 ± 6.28 years; 1.79 ± 0.09 m; 87.92 ± 9.70 Kg)	Lower extremity trauma resulting in unilateral ankle muscle weakness	Customized Nylon 11 PD-AFO (low vs. middle vs. high bending axis)	Kinematics; Kinetics; EMG	Most of the patients (7) preferred the middle bending axis. After EMG test, PD-AFO altered medial gastrocnemius activity in late single-leg support. Low bending axis resulted in the greatest medial gastrocnemius activity. Different bending axis locations had few effects on ankle and knee peak joint kinematics and kinetics.
Sarma et al., 2019 [23]	No data	13% Kevlar Fiber reinforced ultra-high molecular weight polyethylene (UHMWPE)	>1	No data	Customized Kevlar Fiber Reinforced UHMWPE AFO	Kinematics; kinetics; FEM simulations	Based on FEM simulations Kevlar Fiber Reinforced UHMWPE-based composite material was selected as best material for fabrication of AFO compared with ABS, PLA, Nylon 6/6 and PP. The maximum ankle angle during dorsiflexion was 12° and maximum angle during plantar flexion was 23°.
Schrank and Stanhope, 2011 [25]	SLS	Nylon 11 (DuraForm EX Natural Plastic)	2 (1 M; 1 F; 34.50 ± 19.09 years; 1.71 ± 8.49 m; 65.85 ± 8.41 Kg)	Healthy	Customized Nylon 11 PD-AFO	Dimensional accuracy; clinical observation; participant feedback	The dimensional accuracy of the fabricated PD-AFOs was 0.5 mm. The participants demonstrated a fully accommodated, smooth, and rhythmic gait pattern following gait test and reported no discomfort. No signs of uneven pressure distribution, redness, or abrasions.
Telfer et al., 2012 [32]	SLS	Nylon 12 (PA2200)	1 (M, 29 years; 1.85 m; 78.00Kg)	Healthy	Customized Nylon 12 AFO with gas spring vs. Shod only	Kinematics; kinetics	Use of a gas spring to control the stiffness of the AFO. AM AFO led to a lower peak plantarflexion angle at the start stance and higher at the toe-off vs. shod only. Peak ankle internal plantarflexion moment was significantly reduced in both AFO conditions compared to shod. Both AFO conditions also increased peak knee internal flexion moment during the first half of stance. AM AFO clinical performance and biomechanical changes equivalent to TTPP AFO with the advantage of the design freedom provided by AM.
Vasiliauskaite et al., 2019 [31]	SLS	PA12	6 (3M (1 adult, 2 children); 3F (1 adult, 2 children); 23 ± 20 years; 1.5 ± 0.2 m; 52 ± 33 Kg)	1 poly-trauma; 1 Charcot-Marie Tooth; 3 cerebral palsy; 1 bilateral clubfoot	Customized PA12 AFO with carbon strut vs. TTPP AFO vs. Shod Only	Kinematics; kinetics	AM AFO step length significantly increased vs. TTPP AFO due to better energy storage properties. Push-off phase characteristics and joint work in stance became more atypical using AFO and no significant improvements in speed were observed.
Wierzbicka et al., 2017 [27]	FDM	ABS	1 (F; 22 years)	Chronic ankle joint instability	Customized ABS AFO vs. No control	Observation after trial; patient feedback	The AFO was comfortable and fully stabilizing the ankle joint. After gait cycle the test ended with success without no bruises or irritations on patient’s skin. Limitations were found in climbing stairs, riding a bike, and driving a car.

FDM, Fused Deposition Modeling; SLS, Selective Laser Sintering; MJF, Multi-Jet Fusion; SLA, Stereolithography; ABS, Acrylonitrile Butadiene Styrene; TPU, Thermoplastic Polyurethane; PLA, Poly-Lactic Acid; PA12, Polyamide 12; PP, polypropylene; M, Male; F, Female; TTPP, Traditional thermoformed polypropylene; DAFO, Dynamic ankle-foot orthosis; PD-AFO, Passive dynamic ankle-foot orthosis; QUEST, Quebec user evaluation of satisfaction with assistive technology; FEM, finite element model; EMG, electromyography; GRF, Ground reaction force; AM, Additive manufacturing.

**Table 2 bioengineering-09-00249-t002:** GRADE evidence profile.

Quality Assessment							Nº of Patients/Participants		Effect		Quality	Importance
Nº of Studies	Study Design	Risk of Bias	Inconsistency	Indirectness	Imprecision	Other Considerations	Customized AM AFO	Traditional Thermoformed Polypropylene AFO	Relative (95% CI)	Absolute (95% CI)		
**Walking ability through biomechanical tests (kinematics, kinetics, EMG)**												
12	Observational studies [15,22,23,26,28,29,30,31,32,33,34,35]	serious ^a,b^	not serious	Serious ^a^	not serious	none	66 ^g^	9	--	--	⨁◯◯◯ VERY LOW	Important
**Durability through a mechanical test**												
5	Observational studies [15,19,20,21,22]	not serious	not serious	serious ^a,c^	serious ^d^	none	16	2	--	--	⨁◯◯◯ VERY LOW	Important
**Durability through observation after trial**												
2	Observational studies [27,36]	very serious ^e^	not serious	not serious	serious ^d^	none	8	7	--	--	⨁◯◯◯ VERY LOW	Important
**Patient satisfaction assessed with the QUEST**												
2	Observational studies [15,28]	serious ^f^	not serious	not serious	serious ^a,d^	none	2	1	--	--	⨁◯◯◯ VERY LOW	Important
**Comfort through participant/patient feedback**												
6	Observational studies [19,22,24,25,26,27]	very serious ^b,e^	not serious	serious ^a^	serious ^d^	none	17	1	--	--	⨁◯◯◯ VERY LOW	Important
**Dimensional accuracy through FaroArm (fit with a 3 mm spherical tip)**												
1	Observational studies [25]	not serious	not serious	serious ^a^	serious ^d^	none	1	0	--	--	⨁◯◯◯ VERY LOW	Important
**Material strength and AFO behavior simulation assessed by FEM analysis**												
3	Observational studies [19,21,23]	serious ^d^	not serious	serious ^a^	serious ^d^	none	3	1	--	--	⨁◯◯◯ VERY LOW	Important

CI Confidence Interval. ^a^ Not all studies compared to traditionally thermoformed polypropylene AFOs; ^b^ Differences in type of Participants/Patients conditions; ^c^ Differences in type of AM/Traditional AFO assessed; ^d^ Participants/Patients number assessed low; ^e^ No quantitative assessment; ^f^ No blinding of AFOs; ^g^ Sarma et al. [23] does not reference the exact number of participants, so the value of 1 element was considered.

**Table 3 bioengineering-09-00249-t003:** Comparison between the different maximum angles obtained by the ankle and knee of the leg with the AFO at the stance phase.

Reference	N	Healthy/ Unhealthy	Ankle Dorsiflexion (°)	Ankle Plantarflexion (°)	Knee Flexion (°)	Knee Extension (°)
Cha et al., 2017 [15]	1	Unhealthy	22	−8	NA	NA
Liu et al., 2019 [22]	12	Unhealthy	0	−2	13	5
Sarma et al., 2019 [23]	>1	No Data	10	1	NA	NA
Mavroidis et al., 2011 [26]	1	Healthy	15	−8	NA	NA
Chae et al., 2020 [28]	1	Unhealthy	NA	NA	NA	NA
Vasiliauskaite et al., 2019 [31]	6	Unhealthy	13	0.2	12.8	−2
Telfer et al., 2012 [32]	1	Healthy	18 ^1^; 16 ^2^	0 ^1^; −3 ^2^	19 ^1^; 15 ^2^	10 ^1^; 8 ^2^
Lin, Lin, and Chen, 2017 [33]	1	Healthy	NA	NA	20	−1
Choi et al., 2017 [34]	8	Healthy	10	−5	17	5
Harper et al., 2014 [30]	13	Unhealthy	6.55 ^3^; 5.86 ^4^; 5.68 ^5^	−6.59 ^3^; −6.03 ^4^; −5.96 ^5^	13.38 ^3^; 15.71 ^4^; 17.17 ^5^	NA
Creylman et al., 2013 [29]	8	Unhealthy	NA	-3	19	NA
Ranz et al., 2016 [35]	13	Unhealthy	5.83 ^6^; 5.19 ^7^; 4.87 ^8^	−0.68 ^6^; −0.61 ^7^; −0.65 ^8^	17.34 ^6^; 17.46 ^7^; 17.85 ^8^	5.21 ^6^; 4.69 ^7^; 4.91 ^8^

NA: Not Applicable. ^1^ AFO with high stiffness; ^2^ AFO with lowered stiffness; ^3^ AFO stiffness compliant; ^4^ AFO stiffness nominal; ^5^ AFO stiffness stiff; ^6^ AFO with low bending axis; ^7^ AFO with middle bending axis; ^8^ AFO with high bending axis.

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
