# Peer review of "A Review of Additive Manufacturing Studies for Producing Customized Ankle-Foot Orthoses"

_bioengineering, 2022, doi:10.3390/bioengineering9060249_

Round 1

Reviewer 1 Report

1.     Authors mentioned the beginning sentence, “Walking is significant barrier…” This sentence may be remodified with proper adjective.

2.     As per my personal opinion, extraction data information under the heading “material and methods” can be deleted.

3.     Similarly, under the heading of “results” the database collection information the articles considered for the manuscript that include flow diagram is not so significant to the content. Therefore, it can be deleted form the manuscript.

4.      On the whole manuscript, there is no graphical representations mentioned. As a review manuscript it can be added few diagrams based on the references.

Author Response

Dear Reviewer, thank you very much for your insights. We do believe that your expertise provided very good suggestions that allowed us to increase the quality of the manuscript.

  1. Authors mentioned the beginning sentence, “Walking is significant barrier…” This sentence may be remodified with proper adjective.

Dear Reviewer, thank you. We made the amendment according to your suggestion.

  1. As per my personal opinion, extraction data information under the heading “material and methods” can be deleted.

We paid proper attention to your comment. However, that is a requirement for mdpi to review papers. While we do agree that some parts may seem unnecessary, we are keen to follow mdpi instructions. We made some changes, in accordance.

  1. Similarly, under the heading of “results” the database collection information the articles considered for the manuscript that include flow diagram is not so significant to the content. Therefore, it can be deleted form the manuscript.

As previously mentioned, the flow diagram is a request for adequate revision papers, as stated by the editing services of mdpi. Thus, we do think that the diagram should be included in the manuscript.

  1. On the whole manuscript, there is no graphical representations mentioned. As a review manuscript it can be added few diagrams based on the references.

While we do understand your thoughts, we do believe that having the 3 tables is adequate to highlight the results of the manuscript. Indeed, authors should focus on not replicating results on tables, figures, and text.

Reviewer 2 Report

The manuscript entitled “ A review of additive manufacturing studies for producing customized ankle-foot orthoses “ reviews the recent studies using additive manufacturing for the fabrication of Ankle-foot orthoses and the comparison of this technique to the standard thermoplastic vacuum forming in terms of mechanical properties, biomechanics and patients feedback.

The paper is well written, the introduction as well as the Methods are concise. The results and discussion sections are more descriptive of the obtained results. To increase the value of this manuscript, we suggest adding a table summarizing key tests to evaluate 3D printed AFOs based on these papers as well as other “excluded” studies.

1.     The authors mentioned that custom made AFO fabrication takes long time to fabricate, please be more specific and give an average time and compare it to AM techniques such as FDM.

2.     In many passages the word patient was capitalized, is there any raison for that?

3.     Line 152, please give more details on the mechanical properties of AFO made by printed PLA on [19].

4.     Line 174, AFO made by SLS “may show equivalence in clinical performance compared with traditional AFOs”; this statement is contradictor with the mechanical properties of these AFOs where most of them did not survive the test period. Please re-phrase

5.     Line 178, “when an ultraviolet 178 laser contacts the resin [38]” this phrase seems incomplete. Please explain the technology (use of UV and photo initiators). Also, the limitations if the materials used for this technique and the post processing treatments which affect the properties of the final product.

6.     Line 184, “MJF mesh-up SLS and binder jetting’ please rephrase.

7.     Line 192, please include FDM in this comparison.

8.     Line 191, PA12: first time cited abbreviation, please link it to the full name. Please check other abbreviations (QUEST, FDM, SLS, MJF, SLA…) when first appear in the text.

9.     Line 207, how patients’ physiognomies will affect AFO outcomes?

10.  Line 208 to 209, please use lower case for all listed diseases.

11.  Line 223, physiognomy vs anatomy!

Author Response

Dear Reviewer, thank you very much for your insights. We do believe that your expertise provided very good suggestions that allowed us to increase the quality of the manuscript.

The manuscript entitled “ A review of additive manufacturing studies for producing customized ankle-foot orthoses “ reviews the recent studies using additive manufacturing for the fabrication of Ankle-foot orthoses and the comparison of this technique to the standard thermoplastic vacuum forming in terms of mechanical properties, biomechanics and patients feedback.

The paper is well written, the introduction as well as the Methods are concise. The results and discussion sections are more descriptive of the obtained results. To increase the value of this manuscript, we suggest adding a table summarizing key tests to evaluate 3D printed AFOs based on these papers as well as other “excluded” studies.

Dear Reviewer, thank you very much for your suggestions. We have carefully read your comment and that provoked a very good discussion between authors. While we do understand your point of view, we have followed the PRISMA guidelines to make a robust manuscript. The presented tables have the relevant issues to be considered, but if you want to provide more details on your suggestion, we would be more than happy to consider them.

  1. The authors mentioned that custom made AFO fabrication takes long time to fabricate, please be more specific and give an average time and compare it to AM techniques such as FDM.

Line 45 and 224 were re-phrased to be more specific with the time taken to make Custom Made AFO vs AM AFO:

Line 45 However, it takes a long time to make the mold and get the final product, which may take from 2 days to several weeks depending on the post-processing needed. Also, the technician needs to spend most of that time working on the orthosis, taking the time away from the work with patients and other aspects of their work 

Line 224 Currently, the time from the prescription to the design of traditional polypropylene AFO can take several weeks, making them often unusable due to the constant changes in physiognomy, particularly in children. Custom AM AFOs could have an essential role in solving the manufacturing time (less than 1 day), as shown by the two studies using children as participants [15,36].

  1. In many passages the word patient was capitalized, is there any raison for that?

We are sorry for that mistake, as there is no reason at all. We have changed all the words to “patient”

  1. Line 152, please give more details on the mechanical properties of AFO made by printed PLA on [19].

That is a major issue to be discussed in the paper, as followed by our suggestions. Indeed, the quality of the manuscripts is low (as presented) lacking a lot of important pieces of information for further studies, and the advance in science.

Unfortunately, Dal Maso, A.; Cosmi, F. 3D-printed ankle-foot orthosis: A design method. Mater. Today Proc. 2019, 12, 252–261, doi:10.1016/j.matpr.2019.03.122. do not have the mechanical properties described and only say:

ï‚· Material. PLA is an excellent material for FDM, but it is not the most mechanically resistant. Other materials, such as Nylon, could allow thinner profiles maintaining the same mechanical characteristics.

  1. Line 174, AFO made by SLS “may show equivalence in clinical performance compared with traditional AFOs”; this statement is contradictor with the mechanical properties of these AFOs where most of them did not survive the test period. Please re-phrase

We have followed your suggestions, and re-phrased it to “The results suggest that these devices may show equivalence in clinical performance compared with traditional AFOs however, their mechanical performance is far from ideal.”

  1. Line 178, “when an ultraviolet 178 laser contacts the resin [38]” this phrase seems incomplete. Please explain the technology (use of UV and photo initiators). Also, the limitations if the materials used for this technique and the post processing treatments which affect the properties of the final product.

According to your suggestion, we have re-phrased to “SLA is one of the earliest additive manufacturing methods, which was developed in 1986 and uses a liquid-based process that consists of the curing or solidification of a photosensitive polymer when ultraviolet laser contact the resin. SLA prints high-quality parts at a fine resolution as low as 10 μm. However, it is relatively slow and expensive, the range of printing materials is minimal, is sensitive to long exposure to UV light, and the printed parts are affected by moisture, heat, and chemicals.   

  1. Line 184, “MJF mesh-up SLS and binder jetting’ please rephrase.

We have re-phrased to “MJF combine SLS and binder jetting technologies.”

  1. Line 192, please include FDM in this comparison

Dear reviewer, thank you for that insight. We are sure you agree that it is very difficult to compare technologies that use different materials. Anyway, comparing the acquisition costs, we can easily understand the support for “MJF has the lowest cost of 3D printed parts”.

  1. Line 191, PA12: first time cited abbreviation, please link it to the full name. Please check other abbreviations (QUEST, FDM, SLS, MJF, SLA…) when first appear in the text.

We have made a full revision of the paper.

  1. Line 207, how patients’ physiognomies will affect AFO outcomes?

We are sorry for the mistake. The adequate is anatomy, as amended.

  1. Line 208 to 209, please use lower case for all listed diseases.

We have made the amendment as suggested; except Charcot-Marie tooth.

  1. Line 223, physiognomy vs anatomy!

We are sorry for the mistake. The adequate is anatomy, as amended.

Round 2

Reviewer 1 Report

All the necessary comments are made in the manuscript as per the reviewer's comments. Therefore, the manuscript can be published as submitted.